# Enhancing student-centered walking environments on university campuses through street view imagery and machine learning

Yi Qin[1], Xue Wu[2], Tengfei Yu[3], Shuai Jiang[4]*

**1** Department of Art and Craft, Xi'an Academy of Fine Arts, Xi'an, China, **2** Department of Arts, School of Humanities and Social Sciences, Xi'an Jiaotong University, Xi'an, China, **3** School of Arts, Chongqing University, Chongqing, China, **4** College of Architecture and Landscape, Peking University, Beijing, China

* jiangshuai4031@163.com

## Abstract

Campus walking environments significantly influence college students' daily lives and shape their subjective perceptions. However, previous studies have been constrained by limited sample sizes and inefficient, time-consuming methodologies. To address these limitations, we developed a deep learning framework to evaluate campus walking perceptions across four universities in China's Yangtze River Delta region. Utilizing 15,596 Baidu Street View Images (BSVIs), and perceptual ratings from 100 volunteers across four dimensions—aesthetics, security, depression, and vitality—we employed four machine learning models to predict perceptual scores. Our results demonstrate that the Random Forest (RF) model outperformed others in predicting aesthetics, security, and vitality, while linear regression was most effective for depression. Spatial analysis revealed that perceptions of aesthetics, security, and vitality were concentrated in landmark areas and regions with high pedestrian flow. Multiple linear regression analysis indicated that buildings exhibited stronger correlations with depression ($\beta = 0.112$) compared to other perceptual aspects. Moreover, vegetation ($\beta = 0.032$) and meadows ($\beta = 0.176$) elements significantly enhanced aesthetics. This study offers actionable insights for optimizing campus walking environments from a student-centered perspective, emphasizing the importance of spatial design and visual elements in enhancing students' perceptual experiences.

## 1. Introduction

The walking environment in higher education settings plays a critical role in shaping students' daily experiences, influencing their psychological well-being, physical health, and academic performance [1–3]. Despite its significance, many university campuses face challenges related to inadequate walking environments, such as poor spatial design, limited accessibility, and insufficient integration of natural elements. These issues are particularly pressing in China, where the rapid expansion of educational opportunities has led to the proliferation of new campuses over the past two decades [4]. Addressing these challenges is therefore urgent,

**Data availability statement:** All data and code are available from the Figshare database (accession number(s) https://doi.org/10.6084/m9.figshare.28238576.v1)

**Funding:** The author(s) received no specific funding for this work.

**Competing interests:** The authors declare that they have no known competing financial interests or personal relationships that could have appeared to influence the work reported in this paper.

as well-designed walking spaces enriched with natural elements have been shown to positively contribute to students' psychological well-being [5], promote outdoor physical activity [6], and enhance academic performance [7,8].

Walking spaces on campuses encompass pathways that facilitate daily social interactions among students, defined by architectural surroundings and vegetation [9,10]. Environmental psychology views the walking experience as an interplay between the physical landscape and human behavior [11,12], emphasizing the importance of diverse spatial attributes in shaping the subjective perception such as aesthetics, security, depression, and vitality [13–15]. Previous research has examined various dimensions of campus environments, including mental recovery [16], physical activity [17], and academic performance [7]. Specific factors such as the green view index (GVI) [18] and sky view index (SVI) [19] have also been investigated. However, these isolated factors often fall short of capturing the complexity of students' perceptions. Moreover, traditional data collection methods, such as face-to-face interviews and questionnaires [20], are limited by small sample sizes, high costs, and time inefficiency, underscoring the need for more scalable and innovative approaches.

Recent technological advancements have introduced new research possibilities. Crowd-sourced services and Geo-tagged images, such as Baidu Street View images (BSVIs) [21] and Google Street View images [22], provide rich information reflecting street quality. Moreover, the rapid development of computer vision technologies, particularly deep learning and machine learning, has transformed urban and campus planning research [23–25]. These advancements have made it possible to evaluate campus spatial perceptions on a large scale.

This study focuses on quantifying walking perceptions on university campuses, with particular emphasis on four key indicators: aesthetics, security, depression, and vitality. These perceptual dimensions significantly influence pedestrian behaviors and outdoor exercise choices [13–15]. Our primary goal is to identify the visual characteristics associated with these four perceptual indicators and to understand how specific spatial features contribute to distinct walking perceptions. To achieve this, we address the following research questions: 1) how are the four perceptions distributed across each campus? 2) which features influence walking perception and contribute to specific perceptions of a place? 3) how can we efficiently, cost-effectively, and accurately quantify and predict students' campus walking perceptions?

By addressing these questions, our study aims to provide insights into the relationship between walking perception and the physical campus environment, offering practical implications for the design and optimization of campus spaces.

## 2. Literature review

### 2.1. Campus walking space effect on students

Well-designed walking spaces with natural environments on campuses have been shown to significantly impact students' well-being and academic performance [2,12]. These spaces can promote mental health [5], encourage outdoor physical exercise [6], enhance academic performance [26], and help reduce campus violence [27]. Recent studies have further emphasized the importance of campus walking environments in shaping students' perceptions and behaviors. For instance, Ding et al. conducted a systematic review highlighting that active transportation infrastructure, such as increased street connectivity and better walkability, is positively associated with students' physical activity and mental health [28]. Additionally, several studies have identified specific features of walking spaces that significantly influence students' mental health and walking perceptions. For example, the availability of greenery [29], accessibility of facilities [30], proximity to open spaces [31], aesthetic quality [32], and other

walkability-related features have all been shown to correlate strongly with students' psychological well-being and walking behavior. Hipp et al. found that perceived restorative qualities of walking spaces significantly improved students' quality of life, highlighting the importance of designing environments that foster relaxation and mental recovery [33].

Despite the extensive body of research validating the impact of campus walking environments on student well-being, there is a notable gap in accurately identifying and explaining which specific street-view factors influence walking perception in complex campus environments. Traditional investigation methods, while informative, are often costly, time-consuming, and limited in sample size [31]. These limitations necessitate the exploration of new methodologies, such as the use of street-view images and advanced technologies. Recent advancements in urban studies, such as those by Alhassan and Sevtsuk, have demonstrated the potential of spatial analytics and pedestrian modeling to quantify the impact of urban design on social interactions and walking behaviors [34]. These approaches can be adapted to campus environments to provide deeper insights into how specific design elements influence student perceptions and behaviors.

## 2.2.  Measuring walking perception based on street-view images

Street-view images have gained significant traction in urban studies, addressing diverse aspects such as urban greenery [35], health and well-being [36], transportation [37], and walkability [38]. Covering areas inhabited by half of the global population, these images provide a human-centered perspective and are often regarded as a complementary tool to remote sensing imagery [39]. Previous research has leveraged street-view images to measure walking perception at scale by examining physical features such as greenery [40], street canyon enclosure [41], and infrastructure [42]. For instance, Zhou et al. developed a quantified composite walkability index using semantic analysis of street-view images [43]. Furthermore, studies have explored intangible aspects, including urban function inference and perceived streetscape quality [44].

Recent advancements have deepened our understanding of human walking perceptions, encompassing dimensions such as security, comfort, and depression [45,46]. These insights not only inform urban planning and design [47] but also provide practical applications for understanding how walking space features shape human perception [21]. However, research on campus walking perception using street-view images remains limited, with most studies relying on manually collected campus images [4,48,49]. While these efforts have enhanced our understanding of human and student perceptions, challenges persist in extracting high-level information and achieving efficient pixel-wise classification of image features, thereby underscoring the need for more advanced analytical approaches.

## 2.3.  Extracting high-level spatial features using deep learning

The complexity of analyzing urban streets lies in capturing the overall profound features of images rather than merely identifying individual objects [50]. Recent advances in deep learning, machine learning, and computer vision have enabled the accurate, efficient, and automatic extraction of high-level information from large numbers of images [51]. Deep Convolutional Neural Networks (DCNN) have demonstrated remarkable capabilities in automatically extracting image features, achieving significant success in urban science [52] and other domains [53,54]. More advanced algorithms, such as Fully Convolutional Networks (FCN) and SegNet, have further enhanced the extraction of high-level semantic information from images. However, these models often struggle to capture global contextual information effectively [55].

To address this limitation, techniques like Atrous Spatial Pyramid Pooling (ASPP) have been integrated into architectures such as PSPNet [56] and DeepLabV3 [57]. These deep convolutional neural networks with pyramid pooling have shown superior performance in the semantic segmentation of urban street-view images [58, 59]. In this study, we employ DeepLabV3+ [55], an improved version of DeepLabV3, to model students' perceptions and extract high-level semantic information from campus street-view images. This model refines segmentation results by combining low-level and high-level features, enabling more precise boundary delineation. Compared to earlier models, DeepLabV3+ achieves a better balance between computational efficiency and segmentation accuracy, making it particularly suitable for analyzing complex campus environments. This approach aims to overcome the limitations of previous methods and provide a more comprehensive understanding of campus walking perceptions.

## 3. Methods and data

Our research methodology, as illustrated in Fig 1, consisted of three primary stages. First, we extracted road networks from OpenStreetMap (OSM) and generated Geo-tagged sampling points using ArcGIS. Next, we employed the Baidu Street Map API to acquire corresponding BSVIs. In the second stage, we developed an image semantic segmentation neural network using DeepLabV3+ to extract feature percentages from BSVIs. Concurrently, volunteers evaluated the four perceptual dimensions of campus BSVIs samples as training datasets. We then trained different machine learning models to predict perception scores for the remaining images, resulting in spatial perception maps for all four dimensions across campus spaces. Model accuracy was validated using data from four campuses. In the final stage, we conducted statistical analyses, including multiple regression, to identify the contributions of walking space elements to these perceptions. This comprehensive approach enabled us to systematically collect and analyze data, develop predictive models for campus perceptions, and uncover the complex relationships between physical features and perceptual dimensions of campus walking spaces.

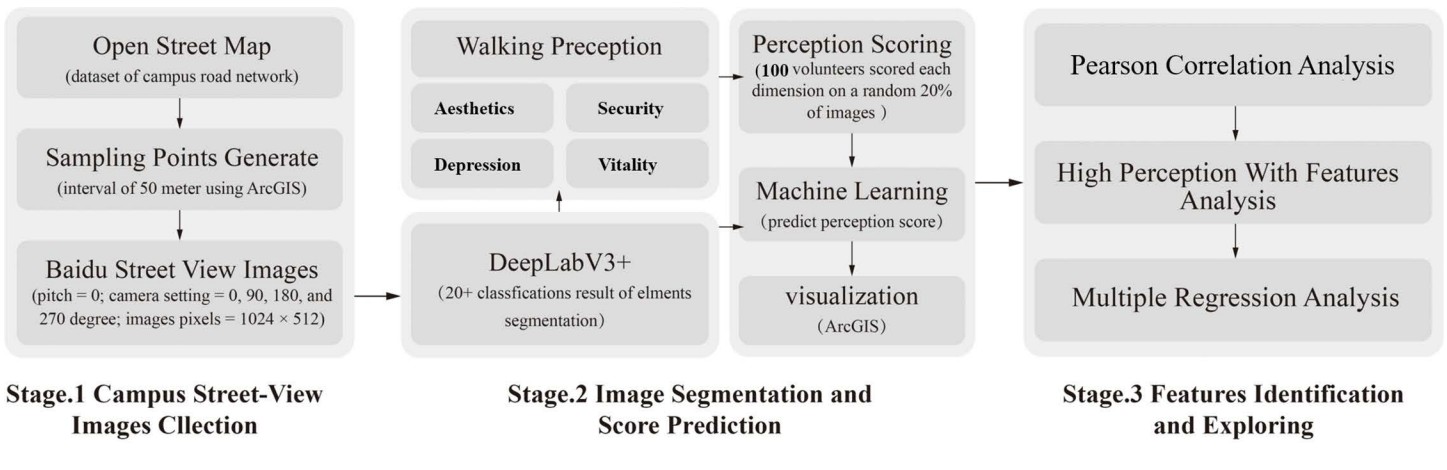

**Fig 1. Research framework.**

### 3.1. Study area

This study focuses on four prominent university campuses in the Yangtze River Delta region of Eastern China (Fig 2): Jiangnan University's Lihu Campus (JNU), Zhejiang University's Zijingang Campus (ZJU), Nanjing University's Xianlin Campus (NJU), and East China Normal University's Minhang Campus (ECNU). Located within the subtropical monsoon zone, these campuses experience distinct seasonal variations, with an average annual temperature of 17.8°C and abundant precipitation. The four campuses share similar climatic conditions, campus sizes, and student populations, as detailed in Table 1. A key criterion for selecting these

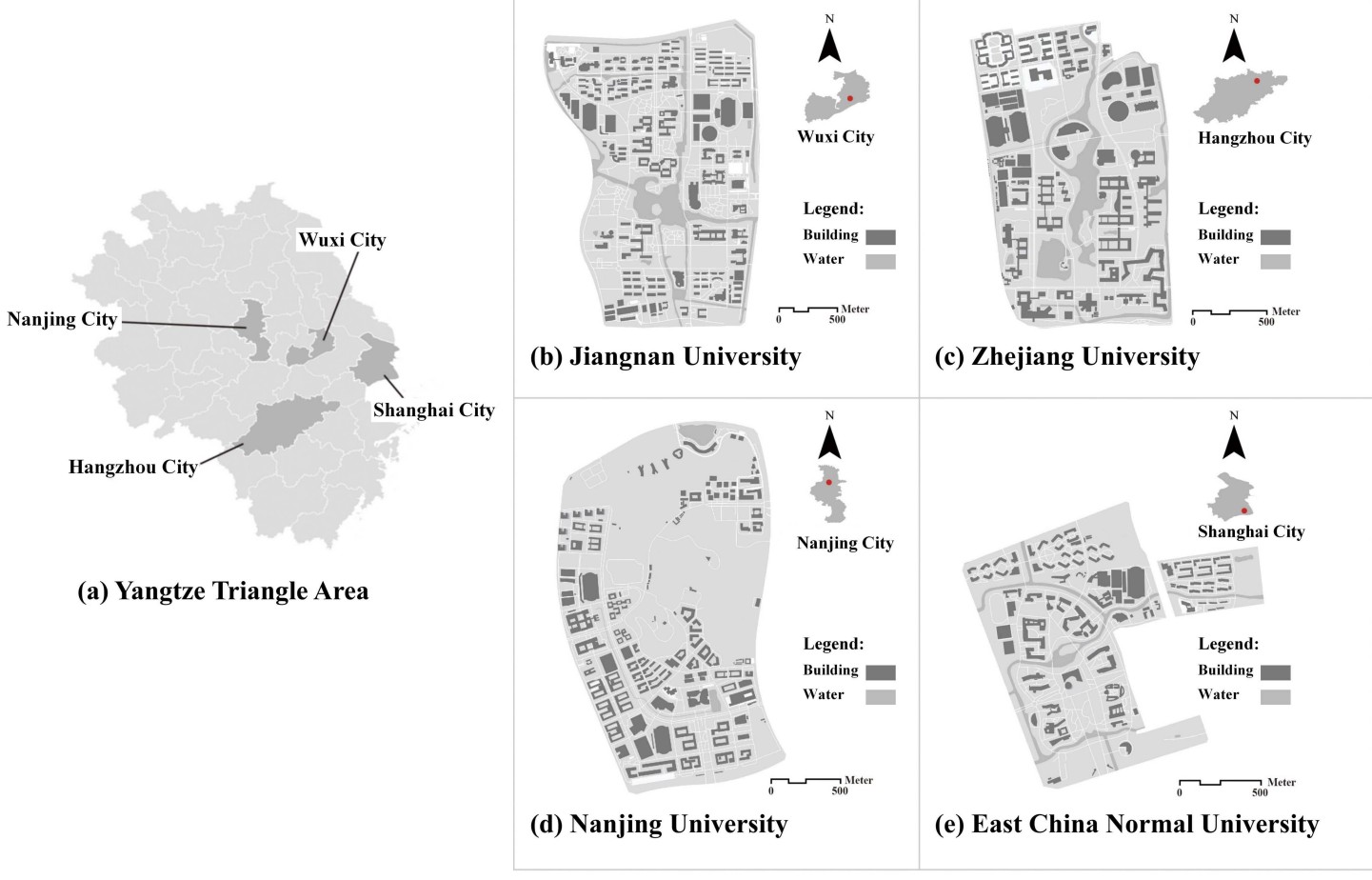

**Fig 2. Overview of the study area for the present study: (a) location of Wuxi, Nanjing, Hangzhou, and Shanghai, (b) Lihu campus of Jiangnan University in Wuxi, (c) Zijingang campus of Zhejiang University, (d) Xianlin campus of Nanjing University, and (e) Minhang campus of East China Normal University.** Basemap from OpenStreetMap (https://www.openstreetmap.org).

**Table 1. The basic situation of the four campuses in this study.**

| Universities | Campus | Province | Floor area (km²) | Students | Completion time |
|---|---|---|---|---|---|
| Jiangnan University (JNU) | Lihu | Jiangsu | 2.16 | 32,219 | 2004.08 |
| Zhejiang University (ZJU) | Zijingang | Zhejiang | 2.12 | 15,867 | 2002.10 |
| Nanjing University (NJU) | Xianlin | Jiangsu | 3.27 | 31,304 | 2009.09 |
| East China Normal University (ECNU) | Minhang | Shanghai | 1.23 | 29,743 | 2006.05 |

campuses was the comprehensive availability of BSVIs. These high-resolution, 360-degree panoramic images provide near-complete visual coverage of each campus, enabling a detailed and consistent analysis of the campus environment across all four campuses.

## 3.2. Street-view images collecting of university campus

Street-view imagery has emerged as a powerful tool in urban science, offering a human-centered, eye-level perspective of pedestrian environments [24,51]. To predict students' perceptions of their surroundings, we utilized the Baidu Map API (https://lbsyun.baidu.com/) to collect street-view data from four university campuses. To ensure comprehensive coverage, we employed ArcGIS software to generate sampling points at 50-meter intervals along campus road networks, sourced from the OpenStreetMap (OSM) platform [60]. This interval was chosen to balance the capture of detailed environmental information with the spatial continuity of visual features [24,61,62].

We standardized image capture parameters for each sampling point: (i) a vertical angle (pitch) set at 0 degrees, (ii) compass headings set at 0, 90, 180, and 270 degrees, and (iii) a maximum image resolution of 1024×512 pixels. These images, captured by Baidu Map between July and September 2017, provide a consistent temporal snapshot of the campus environments (Fig 3). To ensure high data quality, we conducted rigorous data cleaning, removing invalid entries, such as images with poor lighting, suboptimal viewing angles, or duplicates. The final datasets comprises 15,596 valid images, with four directional views from each of the four campuses (JNU, ZJU, NJU, and ECNU). A detailed statistical breakdown of image collection

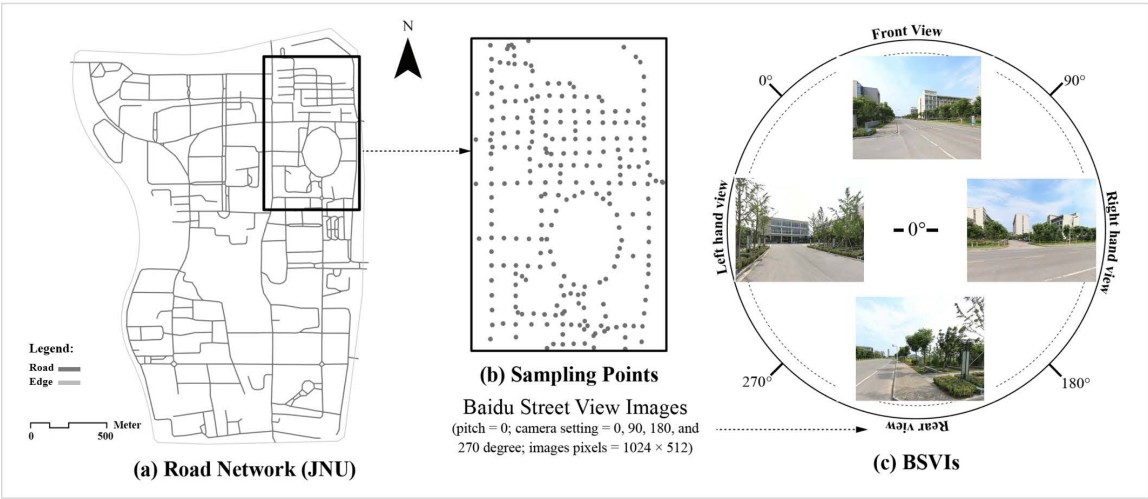

**Fig 3. Example of BSVIs collection process: (a) road network of JNU, (b) generated sampling points interval of 50 meters, and (c) the result of BSVIs collection.** Basemap from OpenStreetMap (https://www.openstreetmap.org); image taken by the author.

**Table 2. Valid street-view image statistics.**

|  | JNU | Ratio | ZJU | Ratio | NJU | Ratio | ECNU | Ratio |
|---|---|---|---|---|---|---|---|---|
| Sampling points | 1,119 |  | 765 |  | 957 |  | 1,125 |  |
| Total images | 4,476 |  | 3,043 |  | 3,828 |  | 4,500 |  |
| Valid images | 4,448 | 99.3% | 2,900 | 95.3% | 3,772 | 98.5% | 4,476 | 99.4% |
| Invalid images | 28 | 0.7% | 143 | 4.7% | 55 | 1.5% | 24 | 0.6% |

for each campus is presented in Table 2. To minimize potential biases in the analysis, we used Python tools to uniformly adjust the brightness, color balance, and dimensions of all BSVIs.

### 3.3. Deep learning-based visual features segmentation

In this study, we employed DeepLabV3+, an advanced deep learning model, to analyze BSVIs. This sophisticated convolutional neural network incorporates Atrous Convolution [57] and Pyramid Pooling, as developed by Google [55]. DeepLabV3+ introduces an innovative decoder module that integrates low-level and high-level features, significantly enhancing segmentation boundary accuracy. Additionally, the model's backbone network was upgraded from ResNet to Xception, enhancing computational efficiency while maintaining superior accuracy compared to other convolutional neural networks.

For model training, we leveraged the Cityscapes datasets, which includes over 30 urban object categories (e.g., cars, roads, sidewalks, buildings, vegetation) sampled from 50 diverse cities [63]. The training process involved 100 epochs on the Cityscapes dataset, achieving a pixel-level accuracy exceeding 89.0% (Fig 4). To quantify walking perception indices, we calculated the proportion of pixels associated with specific features in each image. We then applied a generalized formula (1) to measure various features of campus walking spaces in our BSVIs, enabling a comprehensive analysis of the walking environment.

$$VI_{obj} = \frac{\sum_{i=1}^{n} PIXEL_{obj}}{\sum_{i=1}^{m} PIXEL_{total}}, \; obj \in \{sky, tree, building, etc\} \qquad (1)$$

Here, $VI_{obj}$ is the walking view index, $\sum_{i=1}^{n} PIXEL_{obj}$ is the pixels of walking space feature $obj$, and the $\sum_{i=1}^{m} PIXEL_{total}$ is the total pixels of the street-view images.

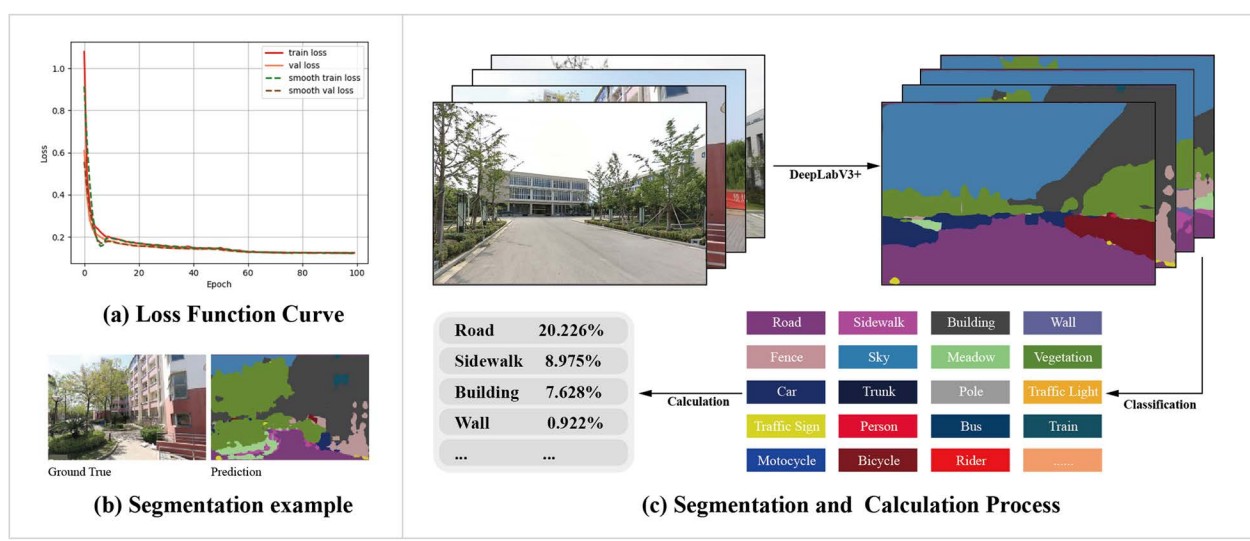

**Fig 4. Segmentation performance based on DeepLabV3+ model and calculation processing: (a) loss function curve in training process based on Cityscapes dataset, (b) example of segmentation results in campus walking space, and (c) calculation process of BSVIs.** Image taken by the author.

## 3.4. Automatic evaluation of the walking perception of student

Drawing inspiration from classical measurement protocols, and recent studies on urban space perception [14,64,65], we focused on four perceptual dimensions: aesthetics, security, depression, and vitality. These qualities are widely recognized in urban design and planning evaluations. Security and aesthetics are particularly crucial for campus life [66], with security playing a pivotal role in preventing incidents such as bullying [67] and criminal activities [68]. Perceived depression serves as a key indicator for investigating the impact of campus spaces on student mental health [69], while vitality has been identified as a critical factor influencing students' movement patterns, daily routines, and outdoor activities [48]. For depression perception, we adopted a momentary measure of environmental depression, which reflects the transient emotional response evoked by the visual characteristics of a space rather than a clinical diagnosis of chronic mental health conditions. This approach aligns with previous studies that have successfully used momentary measures to assess the emotional impact of built environments [14,61]. For detailed questions related to each perceptual indicator, please refer to Table 3.

To ensure robust results, we recruited volunteers from diverse academic backgrounds to evaluate campus walking perceptions using a custom-designed computer platform. Volunteers were required to have at least one year of campus living experience, and to align with the study's target population, we primarily limited the participant pool to university students. Ultimately, 100 volunteers were retained, comprising 57.5% males and 42.5% females, with an average age of 21.3 years. Volunteers rated four perceptual dimensions on a 1–5 scale for a 20% subset (3,119 images) of the total street-view image dataset across four campuses, selected using a stratified random sampling method. Each image was evaluated by at least 50 volunteers, and the mean score was calculated. To ensure the reliability of the results, the standard deviation (std) of the ratings was monitored; if the std exceeded a predefined threshold (e.g., >1.5), additional ratings were collected until the std fell within an acceptable range. This approach minimized potential bias and enhanced the generalizability of the findings. The Ethics Committee of Peking University approved the study. A verbal consent was obtained from the participants of the study.

We employed five machine learning (ML) algorithms - random forest (RF), linear regression (LR), K-nearest neighbor (KNN), and decision tree (DT) - to predict perceptual scores based on the manually rated data and feature proportions obtained from semantic segmentation. These ML models were selected for their proven effectiveness in urban science and architecture. Model performance was evaluated using Mean Absolute Error (MAE) and R-squared [70], calculated as:

$$MAE = \frac{1}{n}\sum_{i=1}^{n}|x_i - x| \qquad (2)$$

**Table 3. The question items utilized to measure four perceptual dimensions.**

| Indicator | Question | Measurement (Likert scale) |
|---|---|---|
| aesthetics | "How visually appealing do you find this campus environment?" | 1-5 |
| security | "How safe do you feel in this campus environment?" | 1-5 |
| depression | "How depressing or gloomy does this campus environment make you feel?" | 1-5 |
| vitality | "How lively or vibrant do you find this campus environment?" | 1-5 |

Note: 1 means strongly disagree, 5 means strongly agree.

where *n* is the number of errors, $x_i$ represents predicted perception scores, and *x* denotes true values. Additionally, we used consistent parameters and reported the average of five training runs as the final outcome.

## 4. Results

### 4.1. Model performances of different models

Table 4 summarizes the prediction performance of four machine learning models across four perceptual dimensions. The evaluation metrics, $R^2$ and MAE, reveal that RF outperformed other models in predicting aesthetics ($R^2$ = 0.81, MAE = 1.69), security ($R^2$ = 0.82, MAE = 1.93), and vitality ($R^2$ = 0.83, MAE = 1.82). In contrast, LR achieved the best performance for depression prediction ($R^2$ = 0.79, MAE = 1.90). Based on these results, RF and LR were selected as the optimal models for predicting perceptual scores across the entire dataset of 15,596 valid campus street-view images.

### 4.2. Overall walking perceptual score of four campuses

The analysis compares the four perceptual scores of JUN campus with those of the other three campuses, as detailed in Table 5. Specifically, JUN campus exhibits a significant higher aesthetics perception score (6.14) compared to ZJU (5.37), NJU (5.84), and ECNU (6.15). In contrast, ECNU achieves the highest scores in both security (6.51) and vitality (6.39), indicating a more secure and vibrant campus environment. However, ZJU has the highest depression score (6.38) compares to other campuses. The standard deviations for each perception score are also provided, reflecting variability in the data across campuses. Pearson correlation analysis result of four perceptual dimensions predicted score among each campus shown in S1 Table.

Fig 5 illustrates the spatial distribution of average walking perception scores across the campus road networks. Aesthetics perception scores are notably higher in areas surrounding landmarks (e.g., libraries, gates) and central locations with high pedestrian activity,

**Table 4. Performance of machine learning models.**

| Model | Aesthetics | | Security | | Depression | | Vitality | |
|---|---|---|---|---|---|---|---|---|
| | $R^2$ | MAE | $R^2$ | MAE | $R^2$ | MAE | $R^2$ | MAE |
| RF | 0.81[*] | 1.69[*] | 0.82[*] | 1.93[*] | 0.86 | 2.19 | 0.83[*] | 1.82[*] |
| LR | 0.79 | 1.89 | 0.74 | 2.27 | 0.79[*] | 1.90[*] | 0.73 | 2.10 |
| KNN | 0.71 | 1.93 | 0.72 | 2.33 | 0.71 | 2.28 | 0.71 | 2.06 |
| DT | 0.78 | 2.01 | 0.75 | 2.30 | 0.66 | 3.12 | 0.73 | 2.30 |

[*]Denotes best performance models for each perception score prediction.

**Table 5. Mean scores, standard deviation of each campus walking perceptions.**

| perceptions | JNU | | ZJU | | NJU | | ECNU | |
|---|---|---|---|---|---|---|---|---|
| | Mean | Std. | Mean | Std. | Mean | Std. | Mean | Std. |
| Aesthetics | **6.41** | 1.27 | 5.37 | 1.50 | 5.84 | 1.25 | 6.15 | 1.49 |
| Security | 5.84 | 1.35 | 5.42 | 1.41 | 5.55 | 1.63 | **6.51** | 1.51 |
| Depression | 5.11 | 1.86 | **6.38** | 1.77 | 5.62 | 1.68 | 4.37 | 2.00 |
| Vitality | 5.24 | 1.36 | 6.00 | 1.70 | 5.11 | 1.59 | **6.39** | 1.36 |

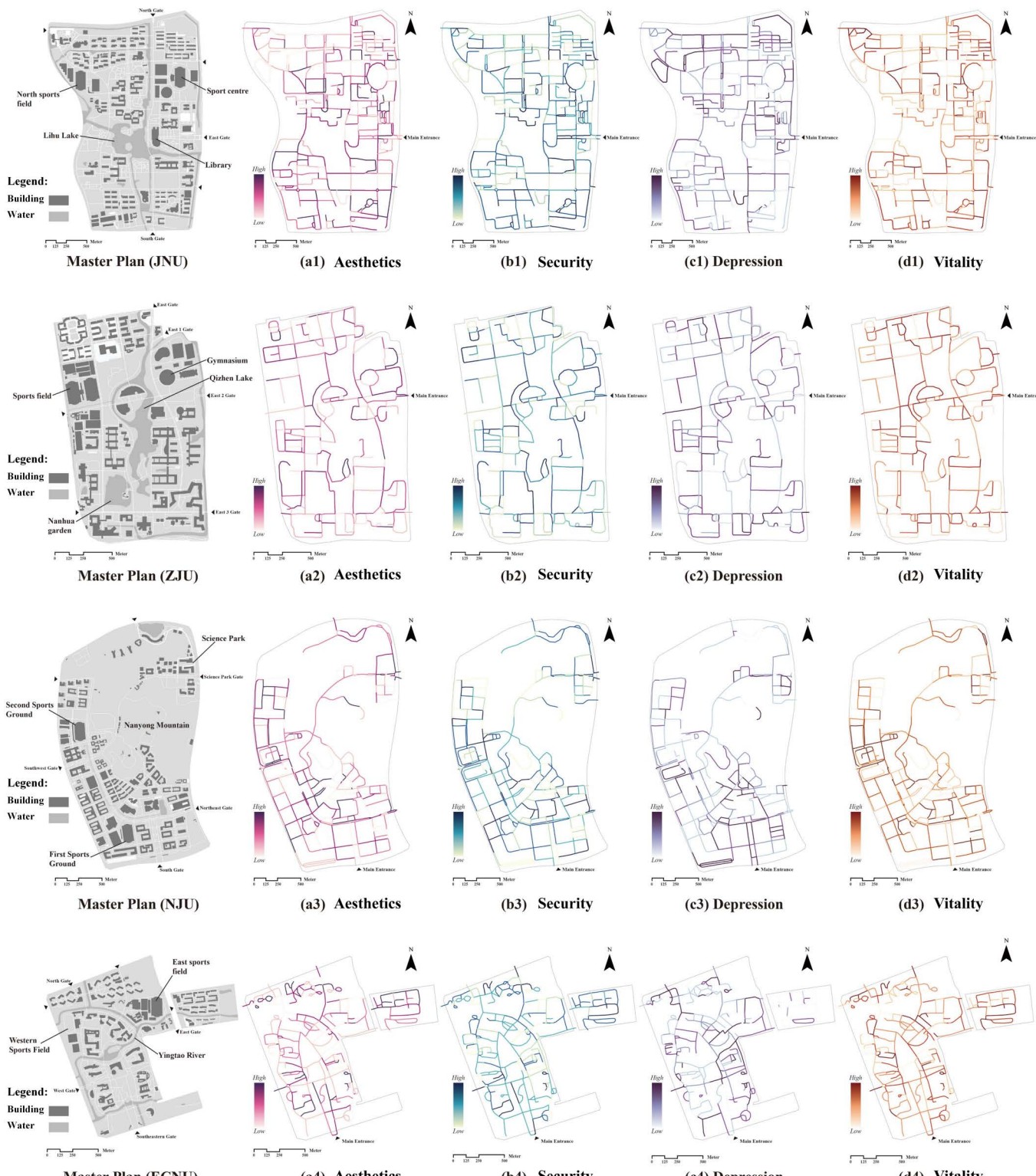

**Fig 5. Analysis of the four dimensions of walking perception on each campus.** Basemap from OpenStreetMap (https://www.openstreetmap.org).

particularly in JUN, ECNU, and ZJU. Additionally, security perception scores are generally high across all four campuses, but lower security scores are observed in peripheral walking spaces, particularly in ZJU (Fig 5b2), indicating potential gaps in security infrastructure such as fencing or surveillance. Depression perception scores are higher in areas with dense building concentrations, as seen in southern JNU (Fig 5c1) and eastern ECNU (Fig 5c4). This observation aligns with prior studies suggesting that densely built environments may limit visibility and reduce sunlight exposure, potentially contributing to stress and psychological discomfort [71].

### 4.3. Features identification and contribution on student perception

**4.3.1. Analysis of the visual features associated with each perceptual.** Our analysis focused on the top 10 most influential visual features, which demonstrated the most significant impact on campus walking perception (Table 6). Notably, vegetation and sky features dominate the visual landscape of on-campus spaces, accounting for 28.82% and 23.59% of all BSVIs, respectively. However, these features also display high variability, with standard deviations ranging from 11.80% to 17.27%. This inconsistency may be attributed to the diverse functional areas within campuses and inter-campus differences. In contrast, walls, cars, and persons constitute a smaller proportion of the visual elements, each accounting for less than 2.00% of the BSVIs. Meadows, sidewalks, and fences occupy an intermediate position, each representing less than 10% of the visual landscape.

**4.3.2. Multiple linear regression analysis of the visual features and walking perception.** To explore the relationships between the top 10 visual features and four walking perceptual dimensions, we conducted a multiple linear regression analysis. Table 7 presents the standardized beta coefficients, highlighting both positive and negative correlations across the dimensions. Person showed the strongest positive correlation with vitality ($\beta = 0.527$, $p < 0.001$) and security ($\beta = 0.236$, $p < 0.001$), while car features exhibited the strongest negative association with aesthetics ($\beta = -0.052$, $p < 0.001$). The meadow ($\beta = 0.176$, $p < 0.001$) and sidewalk ($\beta = 0.154$, $p < 0.001$) features demonstrated substantial positive correlations with aesthetics. Road features consistently showed positive correlations across all dimensions, including aesthetics ($\beta = 0.112$, $p < 0.001$), security ($\beta = 0.136$, $p < 0.001$), and vitality ($\beta = 0.119$, $p < 0.001$). Vegetation showed a positive correlation with other dimensions, albeit with smaller coefficients. Cars showed mixed effects, negatively impacting aesthetics ($\beta = -0.052$, $p < 0.001$) while strongly positively associating with both depression ($\beta = 0.255$, $p < 0.001$) and vitality ($\beta = 0.239$, $p < 0.001$). Buildings demonstrated a stronger positive correlation with

**Table 6. Top 10 visual features identified following segmentation of the BSVIs.**

| Number | Visual features | Mean | Max | Min | Std. |
|---|---|---|---|---|---|
| 1 | Vegetation | 28.82% | 75.51% | 0.00% | 17.27 |
| 2 | Sky | 23.59% | 62.45% | 0.00% | 11.80 |
| 3 | Road | 14.88% | 29.52% | 0.81% | 6.27 |
| 4 | Building | 12.15% | 74.19% | 0.00% | 11.87 |
| 5 | Meadow | 6.02% | 33.32% | 0.00% | 5.07 |
| 6 | Sidewalk | 4.81% | 26.44% | 0.00% | 3.61 |
| 7 | Fence | 4.45% | 58.45% | 0.00% | 5.93 |
| 8 | Wall | 1.84% | 29.30% | 0.00% | 3.42 |
| 9 | Car | 1.37% | 15.81% | 0.00% | 2.07 |
| 10 | Person | 0.70% | 9.99% | 0.00% | 0.99 |

**Table 7. Results of regression analysis for the visual features and perception scores.**

| Standardization features (%) | Aesthetics (*Beta*) | 95% CI | Security (*Beta*) | 95% CI | Depression (*Beta*) | 95% CI | Vitality (*Beta*) | 95% CI |
|---|---|---|---|---|---|---|---|---|
| Vegetation | 0.032*** | [0.031, 0.034] | 0.019*** | [0.017, 0.021] | 0.097*** | [0.096, 0.099] | 0.022*** | [0.021, 0.024] |
| Sky | 0.059*** | [0.057, 0.062] | 0.043*** | [0.040, 0.046] | 0.002 | [−0.000, 0.005] | 0.055*** | [0.052, 0.058] |
| Road | 0.112*** | [0.108, 0.117] | 0.136*** | [0.130, 0.143] | 0.017*** | [0.012, 0.023] | 0.119*** | [0.114, 0.125] |
| Building | 0.013*** | [0.011, 0.015] | 0.044*** | [0.040, 0.047] | 0.112*** | [0.109, 0.115] | 0.038*** | [0.036, −0.042] |
| Meadow | 0.176*** | [0.172, 0.182] | 0.113*** | [0.106, 0.120] | 0.016*** | [0.010, 0.023] | 0.023*** | [0.017, 0.030] |
| Sidewalk | 0.154*** | [0.147, 0.161] | 0.147*** | [0.136, 0.157] | −0.061*** | [−0.070, −0.052] | 0.092*** | [0.083, 0.102] |
| Fence | 0.013*** | [0.009, 0.017] | 0.026*** | [0.021, 0.033] | 0.095*** | [0.090, 0.101] | 0.041*** | [0.035, −0.046] |
| Wall | 0.027*** | [0.019, 0.035] | 0.037* | [0.026, 0.049] | 0.085*** | [0.076, 0.095] | 0.006 | [−0.005, 0.016] |
| Car | −0.052*** | [−0.065, −0.039] | 0.018* | [0.001, 0.037] | 0.255*** | [0.239, 0.271] | 0.239*** | [0.223, 0.257] |
| Person | 0.116*** | [0.091, 0.143] | 0.236*** | [0.199, 0.273] | 0.020 | [−0.012, 0.052] | 0.527*** | [0.493, 0.563] |
| R² | 0.984 | | 0.966 | | 0.970 | | 0.968 | |

Note: 95% CI = 95% confidence interval; Significance levels:

***p < 0.001,

**p < 0.01,

*p < 0.05.

depression (β = 0.112, p < 0.001) compared to other dimensions. Wall features showed non-significant effects on vitality (p > 0.05), while significantly correlating with other dimensions, particularly depression (β = 0.085, p < 0.001).

The results suggest that natural elements, such as meadows and open skies, play a crucial role in enhancing positive perceptions, while built features like walls and dense vegetation may have adverse effects. These findings provide actionable insights for urban planners and designers aiming to create more inviting and psychologically supportive campus environments.

## 5. Discussion

Campus walking space and its perceptions impact the students' mental health and social interaction. Previous studies have highlighted the importance of visual and physical factors in shaping these perceptions. This study makes three key contributions: first, we introduced a new approach combining street-view images and machine learning to comprehensively assess students' walking perceptions across four campuses in China. Second, the relationship between perceptual distribution differences and campus spatial heterogeneity was explored and mapped. Third, the connection between physical features and students' perceptions was identified to directly support campus design practices.

Using street-view images offers a cost-effective method, allowing for the capture of walking perceptions from a human perspective. Unlike satellite images, street-view images enable accurate assessments of features such as the green view index and normalized difference vegetation index through statistical analysis [59]. Our large-scale street-view image analysis not only measured physical attributes accurately, but also predicted intangible perceptions using machine learning. This approach aligns with findings from Zhang et al. and Ramírez et al. [14,72], who demonstrated that intangible perceptions can be effectively measured through such techniques.

Mapping intangible perceptions revealed significant spatial heterogeneity across campus walking environments. Positive perceptions, such as aesthetics, security, and vitality, often

concentrated around campus landmarks, including main gates, libraries, and waterscapes. For instance, natural elements around waterscapes at JNU (Fig 5a1) enhanced the aesthetic quality [73], while open spaces with expansive views and minimal visual obstructions allowed for more sunlight and sky exposure, encouraging prolonged outdoor stays and public activities [74]. Conversely, negative perceptions, such as depression, were more prevalent in peripheral areas and dense building zones, such as dormitories. Notably, administrative, living, and teaching areas exhibited high levels of perceptual agreement, while ecological recreation areas at ZJU showed inconsistent results.

To better understand the relationship between walking space features and perceptual dimensions (aesthetics, security, depression, and vitality), we conducted multiple regression analyses (Section 4.3.2). The top 10 environmental features provide actionable insights for campus design. Specifically, maximizing open spaces with expansive views and minimal visual obstructions (e.g., walls, fences) were linked to higher perceptions of aesthetics and vitality. These areas allow for more sunlight and sky exposure, which can prolong outdoor stays and encourage public activities [4,48]. Campus planners should prioritize the creation and maintenance of such spaces, particularly around key landmarks like libraries and waterscapes, to enhance student well-being and social interaction.

Furthermore, addressing building density is critical. High-density building areas, such as dormitory complexes, were associated with negative perceptions like depression. To mitigate this, future campus designs should incorporate more open spaces, greenery, and natural light in densely built areas. Transitional spaces between buildings, such as courtyards or plazas, can help break up monotony and improve perceptual quality [75]. Additionally, enhancing safety through design is another key consideration. Features like well-lit sidewalks, clear sight lines, and visible landmarks were positively associated with perceptions of security. Campus planners should prioritize these elements, particularly in peripheral or less-trafficked areas, to create a sense of safety and encourage nighttime use of outdoor spaces [76]. By incorporating these strategies, campus designers can create environments that not only support walking but also enhance mental health, social interaction, and overall well-being. These findings provide practical guidance for creating more inclusive, vibrant, and restorative campus spaces.

Our study has three main limitations. First, the findings are based on data from only four campuses, which may constrain the external validity and generalizability of the results. Second, walking perception is influenced by complex factors beyond the four dimensions examined in this study, including individual characteristics and environmental elements not fully captured by street-view images. Additionally, potential biases in the scoring process, stemming from the diverse backgrounds of the volunteers, may have impacted the results. Finally, the reliance on historical street-view images from a specific time period limits our ability to account for seasonal, diurnal, and long-term variations in campus environments and student behaviors. Emerging technologies, such as generative artificial intelligence (Generative AI), offer promising solutions by simulating temporal variations in street-view images, enabling more dynamic analyses of the environmental impacts over time [77].

## 6. Conclusion

Campus walking spaces significantly impact students' mental health and academic performance, yet measuring perceptions on large campuses has been challenging. To address this, we developed a comprehensive framework using street-view images and AI to assess students' walking perceptions across four campuses. Our analysis revealed strong correlations between perception distribution and spatial heterogeneity. For example, JUN achieved the highest

aesthetics perception score (6.14), while ECNU demonstrated stronger agreement in security and vitality perceptions (6.15 and 6.39, respectively). Further, the regression analysis identified key spatial features influencing perceptions. This research provides a cost-effective and scalable framework for evaluating campus walking environments, advancing campus design theory. By demonstrating the feasibility of measuring intangible perceptions through machine learning models, we extend the utility of street-view images in environmental studies. Furthermore, our findings offer actionable insights for designers, highlighting the importance of balancing natural and built environments to enhance student well-being.

## Supporting information

**S1 Table. Pearson correlation of walking perceptions on each campus.**
(DOCX)

## Author contributions

**Conceptualization:** Yi Qin, Shuai Jiang.

**Investigation:** Yi Qin, Xue Wu.

**Methodology:** Yi Qin.

**Project administration:** Shuai Jiang.

**Resources:** Yi Qin.

**Software:** Yi Qin.

**Supervision:** Shuai Jiang.

**Writing – original draft:** Yi Qin.

**Writing – review & editing:** Yi Qin, Xue Wu, Tengfei Yu, Shuai Jiang.

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
