## [Decision Letter · Decision Letter 0]

6 Jan 2025

PONE-D-24-35650Enhancing student-centered walking environments on university campuses through street view imagery and machine learningPLOS ONE

Dear Dr. JIANG,

Thank you for submitting your manuscript to PLOS ONE. After careful consideration, we feel that it has merit but does not fully meet PLOS ONE’s publication criteria as it currently stands. Therefore, we invite you to submit a revised version of the manuscript that addresses the points raised during the review process.

Please carefully consider the comments of the reviewers to comprehensively improve the quality of the manuscript.

We look forward to receiving your revised manuscript.

Kind regards,

Bifeng Zhu

Academic Editor

PLOS ONE

Journal Requirements:

Comments from PLOS Editorial Office: We note that one or more reviewers has recommended that you cite specific previously published works. As always, we recommend that you please review and evaluate the requested works to determine whether they are relevant and should be cited. It is not a requirement to cite these works. We appreciate your attention to this request.

2. Please note that PLOS ONE has spec6ific guidelines on code sharing for submissions in which author-generated code underpins the findings in the manuscript. In these cases, all author-generated code must be made available without restrictions upon publication of the work. Please review our guidelines at https://journals.plos.org/plosone/s/materials-and-software-sharing#loc-sharing-code and ensure that your code is shared in a way that follows best practice and facilitates reproducibility and reuse.

3. In your Methods section, please include additional information about your dataset and ensure that you have included a statement specifying whether the collection and analysis method complied with the terms and conditions for the source of the data.

5. We note that Figures 2, 3, 4 and 5 in your submission contain [map/satellite] images which may be copyrighted. All PLOS content is published under the Creative Commons Attribution License (CC BY 4.0), which means that the manuscript, images, and Supporting Information files will be freely available online, and any third party is permitted to access, download, copy, distribute, and use these materials in any way, even commercially, with proper attribution. For these reasons, we cannot publish previously copyrighted maps or satellite images created using proprietary data, such as Google software (Google Maps, Street View, and Earth). For more information, see our copyright guidelines: http://journals.plos.org/plosone/s/licenses-and-copyright.

a. You may seek permission from the original copyright holder of Figures 2, 3, 4 and 5 to publish the content specifically under the CC BY 4.0 license.  

Additional Comments:

Comments from PLOS Editorial Office: We note that one or more reviewers has recommended that you cite specific previously published works. As always, we recommend that you please review and evaluate the requested works to determine whether they are relevant and should be cited. It is not a requirement to cite these works. We appreciate your attention to this request.

Reviewers' comments:

Reviewer's Responses to Questions

**Comments to the Author**

1. Is the manuscript technically sound, and do the data support the conclusions?

Reviewer #1: Yes

Reviewer #2: Partly

2. Has the statistical analysis been performed appropriately and rigorously? 

Reviewer #1: I Don't Know

Reviewer #2: Yes

3. Have the authors made all data underlying the findings in their manuscript fully available?

Reviewer #1: Yes

Reviewer #2: Yes

4. Is the manuscript presented in an intelligible fashion and written in standard English?

Reviewer #1: Yes

Reviewer #2: Yes

5. Review Comments to the Author

Reviewer #1: Enhancing student-centered walking environments on university campuses through street view imagery and machine learning

The authors' study on enhancing student-centered walking environments through street view imagery and machine learning is highly relevant and insightful. It effectively examines how campus environments impact students' academic performance and mental well-being by developing a deep learning framework. The use of Baidu Street View Images (BSVIs) and machine learning models like Random Forest and linear regression to predict perceptions of aesthetics, security, depression, and vitality offers a fresh approach to understanding campus spaces. The findings, which highlight how different visual elements such as vegetation, sky, and sidewalks influence student perceptions, provide valuable insights into designing more student-friendly walking environments.

This is a particularly interesting study because it links physical and human elements, reflecting the complexity of urban environments. The authors present a robust and consistent analysis. However, I have a few suggestions to further enhance the paper:

• The introduction rightly emphasizes the importance of campus walking perceptions, especially in China, where rapid educational expansion has led to the creation of new campuses. Well-designed walking spaces improve students' psychological health, encourage outdoor physical activity, and enhance academic performance. These spaces are crucial in fostering daily social interactions and are key to improving the overall quality of campus environments, as viewed through environmental psychology.

• In addition to the authors' insights, recent studies have explored how urban design influences social behavior in various dimensions, such as well-being, health, and social relationships. I strongly recommend referring to "How Cities Influence Social Behavior" (MIT Press, 2024), which would provide greater depth and consistency to the arguments in this study.

• The authors' use of recent technological advancements, such as crowdsourced services and geotagged images (Baidu Street View and Google Street View), is a notable strength of the paper. The results are timely and relevant. However, I suggest adding a section in the discussion to explore other emerging technologies. For instance, some recent studies have started evaluating the use of AI in the design and perception of cities, including aesthetics and aspects of well-being (e.g., "Leveraging Generative AI Models in Urban Science"). Expanding the discussion to include these advancements would provide a broader framework for the manuscript and position the research within a more comprehensive technological context.

Reviewer #2: Reviewer Comments to Author

This paper examines the relationship between the environmental features and four dimensions of walking perceptions among Chinese college students across four universities, utilizing street view imagery and machine learning technologies. It is an interesting and important exploratory study with the potential to contribute to the development of more walkable and healthier campus environmental design and planning.

However, several aspects require improvement to enhance the overall quality of the manuscript, particularly in the areas of the literature review, methods, and results. The authors are encouraged to avoid overinterpreting the results and drawing conclusions that extend beyond the scope of the study. Furthermore, the writing somewhat hinders the clarity and readability and requires improvement. Therefore, I recommend a major revision before the manuscript can be considered for publication. Please refer to my specific comments below for more details:

Title.

Please explain why your title is specific to student-centered walking environments, as there are many other campus users such as faculty, staff, and surrounding residents. Is it because you only recruit college students as volunteers? If so, please add more details on the volunteers in the Methods section.

Abstract.

1. The opening sentence should provide a research background that is closely related to the purpose of the study. The current version is too broad and lacks focus.

2. Please include more details about the study methodology. Specifically, mention the number of images analyzed, the involvement of volunteers, and relevant information about the campuses. This will give readers a clearer understanding of the study design and scope.

Introduction.

1. The introduction should clearly state why it is important to study the walking environment in higher education settings? What is the rationale behind focusing on this topic? Highlight the existing problems or challenges posed by inadequate walking environments and their impacts on students' daily academic experiences or health.

2. Consider combining paragraphs 2 and 3 to create a more cohesive and organized transition

Literature review

1. Consider incorporating more relevant, recent literature on each subtopic. For example, in section 2.1, as your study focuses on walking environments in university campuses and college students’ perception, please include references that specifically address these aspects. Please refer to the systematic review paper: Ding, Y., Lee, C., Chen, X., Song, Y., Newman, G., Lee, R., ... & Sohn, W. (2024). Exploring the association between campus environment of higher education and student health: A systematic review of findings and measures. Urban forestry & urban greening, 91, 128168.

2. Lines 122–124: Provide references to more recent literature to ensure the review remains current and reflective of recent advances in the field.

3. Lines 156–157: Please specify why DeepLabV3+ was chosen for this study. Include justification for its suitability compared to other methods. In addition, fix the grammatical issues in this sentence to ensure clarity and correctness.

Methods.

1. Line 164, why did you choose 50-meter intervals? Provide supporting evidence, references, or a rationale to justify this decision.

2. Line 168, please provide additional details about the volunteers. Are they college students, professionals, or experts in the field? What qualifications make them suitable for image evaluations? Were there any inclusion or exclusion criteria.

3. Lines 211-212, please specify how you cleaned up the data and excluded invalid images. What criteria were used to determine invalid images, and what steps were taken to ensure data quality and consistency during preprocessing?

4. Lines 256-257, is a participant pool of 50 individuals sufficient to generalize the perception scores across a dataset of 15,596 images, particularly considering the dataset likely includes diverse images with varying levels of environmental perceptions? Address potential limitations due to the small participant pool and justify how you ensure the perception scores are not biased.

5. Lines 259, please provide more details on the question items utilized to measure four perceptual dimensions. In addition, regarding the measurement of depression, this term often refers to a chronic mental health condition, and I was concerned about its validity as a momentary measure. Please conduct further research on depression measurement to ensure the measurement approach is justified and appropriate.

Results.

1. Lines 277-279, please justify this sentence by providing valid and relevant references to support the claim.

2. Lines 327-329, please specify how perception scores were linked to each feature to create Figure 7. From the methods described, it seems volunteers evaluated perception scores for each image rather than for individual features. If this is the case, explain the process used to associate perception scores with specific features.

3. Line 335-337, consider moving this sentence to the Discussion section to maintain focus in the Results section on interpreting data.

4. For table 7, please refine the table following the APA format, including key statistical information such as 95% confidence intervals and R-squared values. Also specify the units of the measurement for the 10 features in this analysis.

5. Table 7 indicates that the vegetation was negatively associated with aesthetics, security and vitality, and positively related to depression. This is kind of conflating with previous studies. The interpretation of these results (Lines 407–409) appears abrupt, ambiguous, and overly intuitive. How can it be determined that vegetation is "excessive and dense" rather than "well-maintained and healthy"? Provide evidence or further analysis to support this statement. If this claim is speculative, acknowledge the uncertainty and suggest possible explanations in the Discussion section.

Discussion/Conclusion.

1. For limitations, it is important to highlight that this is an exploratory study based on data from only four schools, which may limit external validity and generalizability.

2. Ensure that the discussion and conclusion are closely aligned with the study’s actual findings. Please avoid over interpretation (i.e. discussion about excessive vegetation) and ambitious conclusions related to theory development (Line 430).

3. Expand on the implications for future research and campus design guidelines. Discuss how these findings can inform practical strategies for creating more walkable and health-promoting campuses.

Other issues

1. Please proofread the manuscript thoroughly to correct all grammatical errors and typos, enhancing its readability.

6. PLOS authors have the option to publish the peer review history of their article (what does this mean? ). If published, this will include your full peer review and any attached files.

**Do you want your identity to be public for this peer review?** For information about this choice, including consent withdrawal, please see our Privacy Policy .

Reviewer #1: No

Reviewer #2: No

---

## [Author Response · Author response to Decision Letter 1]

28 Jan 2025

Reviewer 1

Reviewer #1: Enhancing student-centered walking environments on university campuses through street view imagery and machine learning

The authors' study on enhancing student-centered walking environments through street view imagery and machine learning is highly relevant and insightful. It effectively examines how campus environments impact students' academic performance and mental well-being by developing a deep learning framework. The use of Baidu Street View Images (BSVIs) and machine learning models like Random Forest and linear regression to predict perceptions of aesthetics, security, depression, and vitality offers a fresh approach to understanding campus spaces. The findings, which highlight how different visual elements such as vegetation, sky, and sidewalks influence student perceptions, provide valuable insights into designing more student-friendly walking environments.

This is a particularly interesting study because it links physical and human elements, reflecting the complexity of urban environments. The authors present a robust and consistent analysis. However, I have a few suggestions to further enhance the paper:

• The introduction rightly emphasizes the importance of campus walking perceptions, especially in China, where rapid educational expansion has led to the creation of new campuses. Well-designed walking spaces improve students' psychological health, encourage outdoor physical activity, and enhance academic performance. These spaces are crucial in fostering daily social interactions and are key to improving the overall quality of campus environments, as viewed through environmental psychology.

> Thank you for your comments.

• In addition to the authors' insights, recent studies have explored how urban design influences social behavior in various dimensions, such as well-being, health, and social relationships. I strongly recommend referring to "How Cities Influence Social Behavior" (MIT Press, 2024), which would provide greater depth and consistency to the arguments in this study.

> Thank you for your suggestion. We have added this content in the latest version. Please refer to Lines 96-114.

• The authors' use of recent technological advancements, such as crowdsourced services and geotagged images (Baidu Street View and Google Street View), is a notable strength of the paper. The results are timely and relevant. However, I suggest adding a section in the discussion to explore other emerging technologies. For instance, some recent studies have started evaluating the use of AI in the design and perception of cities, including aesthetics and aspects of well-being (e.g., "Leveraging Generative AI Models in Urban Science"). Expanding the discussion to include these advancements would provide a broader framework for the manuscript and position the research within a more comprehensive technological context.

> Thank you for your comments. We have incorporated a discussion on emerging technologies in the latest version. Please refer to Lines 439-442.

> Revised (Lines 439-442): Emerging technologies, such as generative artificial intelligence (Generative AI), offer promising solutions by simulating temporal variations in street-view images, enabling more dynamic analyses of environmental impacts over time (Liu et al., 2024).

Reviewer 2

Reviewer #2: Reviewer Comments to Author

This paper examines the relationship between the environmental features and four dimensions of walking perceptions among Chinese college students across four universities, utilizing street view imagery and machine learning technologies. It is an interesting and important exploratory study with the potential to contribute to the development of more walkable and healthier campus environmental design and planning.

However, several aspects require improvement to enhance the overall quality of the manuscript, particularly in the areas of the literature review, methods, and results. The authors are encouraged to avoid overinterpreting the results and drawing conclusions that extend beyond the scope of the study. Furthermore, the writing somewhat hinders the clarity and readability and requires improvement. Therefore, I recommend a major revision before the manuscript can be considered for publication. Please refer to my specific comments below for more details:

Title.

Please explain why your title is specific to student-centered walking environments, as there are many other campus users such as faculty, staff, and surrounding residents. Is it because you only recruit college students as volunteers? If so, please add more details on the volunteers in the Methods section.

> Thank you for raising this question. Our research is more focused on the student population. We have included additional information about the volunteers in the latest version. Please refer to Lines 281-286.

> Revised (Lines 281-286): To ensure robust results, we recruited volunteers from diverse academic backgrounds to evaluate campus walking perceptions using a custom-designed computer platform. Volunteers were required to have at least one year of campus life experience, and to align with the study's target population, we primarily limited the participant pool to university students. Ultimately, 100 volunteers were retained, with 57.5% male and an average age of 21.30.

Abstract.

1. The opening sentence should provide a research background that is closely related to the purpose of the study. The current version is too broad and lacks focus.

2. Please include more details about the study methodology. Specifically, mention the number of images analyzed, the involvement of volunteers, and relevant information about the campuses. This will give readers a clearer understanding of the study design and scope.

> Response to Q1 and Q2: We greatly appreciate your feedback. In the latest version, we have refined the abstract section to provide more details about our study.

> Revised (Lines 19-38): Campus walking environments significantly influence college students' daily lives and shape their subjective perceptions. However, previous studies have been constrained by limited sample sizes and inefficient, time-consuming methodologies. To address these limitations, we developed a deep learning framework to evaluate campus walking perceptions across four universities in China's Yangtze River Delta region. Utilizing 15,596 Baidu Street View Images (BSVIs) and perceptual ratings from 100 volunteers across four dimensions—aesthetics, security, depression, and vitality—we employed four machine learning models to predict perceptual scores. Our results demonstrate that the Random Forest (RF) model outperformed others in predicting aesthetics, security, and vitality, while linear regression was most effective for depression. Spatial analysis revealed that perceptions of aesthetics, security, and vitality were concentrated in landmark areas and regions with high pedestrian flow. Multiple linear regression analysis indicated that buildings exhibited stronger correlations with depression (β = 0.112) compared to other perceptual aspects. Conversely, vegetation (β = 0.032) and meadow (β = 0.176) elements significantly enhanced aesthetics. This study offers actionable insights for optimizing campus walking environments from a student-centered perspective, emphasizing the importance of spatial design and visual elements in enhancing students' perceptual experiences.

Introduction.

1. The introduction should clearly state why it is important to study the walking environment in higher education settings? What is the rationale behind focusing on this topic? Highlight the existing problems or challenges posed by inadequate walking environments and their impacts on students' daily academic experiences or health.

2. Consider combining paragraphs 2 and 3 to create a more cohesive and organized transition

> Response to Q1 and Q2: We sincerely appreciate your comments. We have thoroughly revised the introduction section.

> Revised (Lines 43-71): The walking environment in higher education settings plays a critical role in shaping students' daily experiences, influencing their psychological well-being, physical health, and academic performance (De Vos et al., 2023; Gomez et al., 2021; Lee et al., 2020). Despite its significance, many university campuses face challenges related to inadequate walking environments, such as poor spatial design, limited accessibility, and insufficient integration of natural elements. These issues are particularly pressing in China, where the rapid expansion of educational opportunities has led to the proliferation of new campuses over the past two decades (Wang et al., 2021). Addressing these challenges is therefore urgent, as well-designed walking spaces enriched with natural elements have been shown to positively contribute to students' psychological well-being (Wang et al., 2018), promote outdoor physical activity (Dyment et al., 2007), and enhance academic performance (Hodson et al., 2017; Kweon et al., 2017).

Walking spaces on campuses encompass pathways that facilitate daily social interactions among students, defined by architectural surroundings and vegetation (Abusaada et al., 2021; Balsa-Barreiro et al., 2024). Environmental psychology views the walking experience as an interplay between the physical landscape and human behavior (Ewing et al., 2009; Zhang, Wang, et al., 2024), emphasizing the importance of diverse spatial attributes in shaping the subjective perception such as aesthetics, security, depression, and vitality (Dubey et al., 2016; Li et al., 2022; Zhang et al., 2018). Previous research has examined various dimensions of campus environments, including mental recovery (Akpinar, 2016), physical activity (Mårtensson et al., 2014), and academic performance (Hodson & Sander, 2017). Specific factors such as the green view index (GVI) (Yang et al., 2020) and sky view index (SVI) (Xi et al., 2012) have also been investigated. However, these isolated factors often fall short of capturing the complexity of students' perceptions. Moreover, traditional data collection methods, such as face-to-face interviews and questionnaires (Montello et al., 2017), are limited by small sample sizes, high costs, and time inefficiency, underscoring the need for more scalable and innovative approaches.

Literature review

1. Consider incorporating more relevant, recent literature on each subtopic. For example, in section 2.1, as your study focuses on walking environments in university campuses and college students’ perception, please include references that specifically address these aspects. Please refer to the systematic review paper: Ding, Y., Lee, C., Chen, X., Song, Y., Newman, G., Lee, R., ... & Sohn, W. (2024). Exploring the association between campus environment of higher education and student health: A systematic review of findings and measures. Urban forestry & urban greening, 91, 128168.

2. Lines 122–124: Provide references to more recent literature to ensure the review remains current and reflective of recent advances in the field.

> Response to Q1 and Q2: We sincerely appreciate your comments. In the latest version, we have reorganized the literature review section to provide a more comprehensive understanding of the campus walking environment.

> Revised (Lines 96-114): Well-designed walking spaces with natural environments on campuses have been shown to significantly impact students' well-being and academic performance (Lee & Shepley, 2020; Zhang, Wang, et al., 2024). These spaces can promote mental health (Wang et al., 2018), encourage outdoor physical exercise (Dyment & Bell, 2007), enhance academic performance (Li et al., 2016), and reduce campus violence (Fernandez, 2005). Recent studies have further emphasized the importance of campus walking environments in shaping students' perceptions and behaviors. For instance, Ding et al. (2024) conducted a systematic review highlighting that active transportation infrastructure, such as increased street connectivity and better walkability, is positively associated with students' physical activity and mental health. Additionally, several studies have identified specific features of walking spaces that significantly influence students' mental health and walkability perceptions. For example, the availability of greenery (De Vries et al., 2013), accessibility of facilities (Leyden et al., 2024), proximity to open spaces (Zhang, Sun, et al., 2024), aesthetic quality (Y. Zhao et al., 2017), and other walkability-related features have been shown to correlate strongly with students' psychological well-being and walking behavior. Hipp et al. (2016) found that perceived restorative qualities of walking spaces significantly improved students' quality of life, highlighting the importance of designing environments that foster relaxation and mental recovery.

> Revised (Lines 141-151): Recent advancements have deepened our understanding of human walking perceptions, encompassing dimensions such as security, comfort, and depression (Liu et al., 2021; Min et al., 2020). These insights not only inform urban planning and design (Shen et al., 2018) but also provide practical applications for understanding how walking space features shape human perception (Wang et al., 2022). However, research on campus walking perception using street-view images remains limited, with most studies relying on manually collected campus images (Q. Y. Liu et al., 2022; Tudorie et al., 2020; Wang et al., 2021). While these efforts have enhanced our understanding of human and student perceptions, challenges persist in extracting high-level information and achieving efficient pixel-wise classification of image features, underscoring the need for more advanced analytical approaches.

3. Lines 156–157: Please specify why DeepLabV3+ was chosen for this study. Include justification for its suitability compared to other methods. In addition, fix the grammatical issues in this sentence to ensure clarity and correctness.

> Thank you very much for your comments. We have added further justification for selecting DeepLabV3+ as the foundational model.

> Revised (Lines 164-176): To address this limitation, techniques like Atrous Spatial Pyramid Pooling (ASPP) have been integrated into architectures such as PSPNet (H. Zhao et al., 2017) and DeepLabV3 (Chen et al., 2017). These deep convolutional neural networks with pyramid pooling have shown superior performance in the semantic segmentation of urban street-view images (Gong et al., 2019; Tong et al., 2020). In this study, we employ DeepLabV3+ (Chen et al., 2018), an improved version of DeepLabV3, to model students' perceptions and extract high-level semantic information from campus street-view images. This model refines segmentation results by combining low-level and high-level features, enabling more precise boundary delineation. Compared to earlier models, DeepLabV3+ achieves a better balance between computational efficiency and segmentation accuracy, making it particularly suitable for analyzing complex campus environments. This approach aims to overcome the limitations of previous methods and provide a more comprehensive understanding of campus walking perceptions.

Methods.

1. Line 164, why did you choose 50-meter intervals? Provide supporting evidence, references, or a rationale to justify this decision.

> Thank you very much for your comments. A 50-meter interval allows for a more comprehensive capture of environmental information and ensures the continuity of visual features at a spatial scale.

> Revised (Lines 218-222): To ensure comprehensive coverage, we employed ArcGIS software to generate sampling points at 50-meter intervals along campus road networks, sourced from the OpenStreetMap (OSM) platform (Haklay et al., 2008). This interval was chosen to balance the capture of detailed environmental information with the spatial continuity of visual features (Ito et al., 2024; Ma et al., 2024; Zhao et al., 2023).

2. Line 168, please provide additional details about the volunteers. Are they college students, professionals, or experts in the field? What qualifications make them suitable for image evaluations? Were there any inclusion or exclusion criteria.

> Thank you v

---

## [Decision Letter · Decision Letter 1]

25 Feb 2025

PONE-D-24-35650R1Enhancing student-centered walking environments on university campuses through street view imagery and machine learningPLOS ONE

Dear Dr. JIANG,

Thank you for submitting your manuscript to PLOS ONE. After careful consideration, we feel that it has merit but does not fully meet PLOS ONE’s publication criteria as it currently stands. Therefore, we invite you to submit a revised version of the manuscript that addresses the points raised during the review process.

Please pay attention to the pixels in the figures, so that it can present the information clearly and completely.Please proofread the language to ensure there are no errors.

We look forward to receiving your revised manuscript.

Kind regards,

Bifeng Zhu

Academic Editor

PLOS ONE

Journal Requirements:

Reviewers' comments:

Reviewer's Responses to Questions

**Comments to the Author**

1. If the authors have adequately addressed your comments raised in a previous round of review and you feel that this manuscript is now acceptable for publication, you may indicate that here to bypass the “Comments to the Author” section, enter your conflict of interest statement in the “Confidential to Editor” section, and submit your "Accept" recommendation.

Reviewer #1: All comments have been addressed

Reviewer #2: All comments have been addressed

2. Is the manuscript technically sound, and do the data support the conclusions?

Reviewer #1: Yes

Reviewer #2: Yes

3. Has the statistical analysis been performed appropriately and rigorously? 

Reviewer #1: Yes

Reviewer #2: Yes

4. Have the authors made all data underlying the findings in their manuscript fully available?

Reviewer #1: Yes

Reviewer #2: Yes

5. Is the manuscript presented in an intelligible fashion and written in standard English?

Reviewer #1: Yes

Reviewer #2: Yes

6. Review Comments to the Author

Reviewer #1: Dear Authors,

I would like to commend you for the comprehensive and thoughtful revisions made to your manuscript. Your responses effectively addressed all of my concerns, and the improvements have enhanced the overall clarity and impact of your work.

Given these significant enhancements, I am pleased to recommend your manuscript for publication. Congratulations on this achievement, and I wish you success in your future research endeavors.

Best regards, the Ass. Editor

Reviewer #2: The authors have effectively addressed all reviewer comments and revised the paper in line with the suggestions. The new version has improved significantly in both presentation and content, now appearing much more incisive overall.

Some minor issues need to be addressed before publication:

1. Replace “Person correlation” with “Pearson correlation” in Figure 1.

2. Clarify where Appendix Table S1 is cited in the main text.

7. PLOS authors have the option to publish the peer review history of their article (what does this mean? ). If published, this will include your full peer review and any attached files.

**Do you want your identity to be public for this peer review?** For information about this choice, including consent withdrawal, please see our Privacy Policy .

Reviewer #1: No

Reviewer #2: No

---

## [Author Response · Author response to Decision Letter 2]

26 Feb 2025

Reviewer 1

Reviewer #1: Dear Authors,

I would like to commend you for the comprehensive and thoughtful revisions made to your manuscript. Your responses effectively addressed all of my concerns, and the improvements have enhanced the overall clarity and impact of your work.

Given these significant enhancements, I am pleased to recommend your manuscript for publication. Congratulations on this achievement, and I wish you success in your future research endeavors.

Best regards, the Ass. Editor

> Thank you very much for your review comments.

Reviewer 2

Reviewer #2: The authors have effectively addressed all reviewer comments and revised the paper in line with the suggestions. The new version has improved significantly in both presentation and content, now appearing much more incisive overall.

Some minor issues need to be addressed before publication:

1. Replace “Person correlation” with “Pearson correlation” in Figure 1.

> Thank you for your comments. We have fixed this error, please see Line 189.

2. Clarify where Appendix Table S1 is cited in the main text.

> Thank you for your comments. We have revised this problem.

> Revised (Lines 324-326): Pearson correlation analysis result of four perceptual dimensions predicted score among each campus shown in Appendix Table S1.

---

## [Editor Report · Decision Letter 2]

28 Feb 2025

Enhancing student-centered walking environments on university campuses through street view imagery and machine learning

PONE-D-24-35650R2

Dear Dr. JIANG,

We’re pleased to inform you that your manuscript has been judged scientifically suitable for publication and will be formally accepted for publication once it meets all outstanding technical requirements.

Kind regards,

Bifeng Zhu

Academic Editor

PLOS ONE
---

## [Editor Report · Acceptance letter]

PONE-D-24-35650R2

PLOS ONE

Dear Dr. JIANG,

I'm pleased to inform you that your manuscript has been deemed suitable for publication in PLOS ONE. Congratulations! Your manuscript is now being handed over to our production team.

Kind regards,

on behalf of

Dr. Bifeng Zhu

Academic Editor

PLOS ONE